# Gene Coexpression Network Analysis Indicates that Hub Genes Related to Photosynthesis and Starch Synthesis Modulate Salt Stress Tolerance in *Ulmus pumila*

**DOI:** 10.3390/ijms22094410

**Published:** 2021-04-23

**Authors:** Panfei Chen, Peng Liu, Quanfeng Zhang, Chenhao Bu, Chunhao Lu, Sudhakar Srivastava, Deqiang Zhang, Yuepeng Song

**Affiliations:** 1Beijing Advanced Innovation Center for Tree Breeding by Molecular Design, Beijing Forestry University, Beijing 100083, China; panfeichen@bjfu.edu.cn (P.C.); ackliup@163.com (P.L.); buchenhao@foxmail.com (C.B.); chunhao_lu@163.com (C.L.); srivastavasudhakar@gmail.com (S.S.); DeqiangZhang@bjfu.edu.cn (D.Z.); 2National Engineering Laboratory for Tree Breeding, College of Biological Sciences and Technology, Beijing Forestry University, No. 35, Qinghua East Road, Beijing 100083, China; 3Key Laboratory of Genetics and Breeding in Forest Trees and Ornamental Plants, College of Biological Sciences and Technology, Beijing Forestry University, No. 35, Qinghua East Road, Beijing 100083, China; 4Hebei Academy of Forestry and Grassland Sicences, No. 75, Xuefu Road, Shijiazhuang 050061, China; zqfeng10@163.com

**Keywords:** *Ulmus pumila* L., salt treatment, transcriptome, photosynthesis, starch synthesis

## Abstract

*Ulmus pumila* L. is an excellent afforestation and biofuel tree that produces high-quality wood, rich in starch. In addition, *U. pumila* is highly adaptable to adverse environmental conditions, which is conducive to its utilization for vegetating saline soils. However, little is known about the physiological responses and transcriptional regulatory network of *U. pumila* under salt stress. In this study, we exposed five main cultivars in saline–alkali land (Upu2, 5, 8, 11, and 12) to NaCl stress. Of the five cultivars assessed, Upu11 exhibited the highest salt resistance. Growth and biomass accumulation in Upu11 were promoted under low salt concentrations (<150 mM). However, after 3 months of continuous treatment with 150 mM NaCl, growth was inhibited, and photosynthesis declined. A transcriptome analysis conducted after 3 months of treatment detected 7009 differentially expressed unigenes (DEGs). The gene annotation indicated that these DEGs were mainly related to photosynthesis and carbon metabolism. Furthermore, PHOTOSYNTHETIC ELECTRON TRANSFERH (UpPETH), an important electron transporter in the photosynthetic electron transport chain, and UpWAXY, a key gene controlling amylose synthesis in the starch synthesis pathway, were identified as hub genes in the gene coexpression network. We identified 25 and 62 unigenes that may interact with PETH and WAXY, respectively. Overexpression of UpPETH and UpWAXY significantly increased the survival rates, net photosynthetic rates, biomass, and starch content of transgenic *Arabidopsis* plants under salt stress. Our findings clarify the physiological and transcriptional regulators that promote or inhibit growth under environmental stress. The identification of salt-responsive hub genes directly responsible for photosynthesis and starch synthesis or metabolism will provide targets for future genetic improvements.

## 1. Introduction

Elm (*Ulmus pumila* L.) occurs naturally from the northern cold temperate zone to the subtropics, and in East Asia is used mainly for timber, shelter, food, medicine, fodder, and ecological protection [1,2,3,4]. The wood of *U. pumila* is durable with a clear texture, moderate strength, and good rot resistance. These properties make it suitable for ship building, architecture, furniture, decoration, and handicrafts [5,6,7,8]. *U. pumila* has a long history of cultivation, and it has been used as a life-saving tree during famine due to the edible and medicinal properties of its bark, roots, leaves, and fruits [9,10]. The bark is rich in sterols and starch, with up to 16.14% fiber and 17% polysaccharides, and is used for food, paper, and other products [11,12]. The seeds are edible and contain as much as 25.5% oil; they can be pressed for soap-making and other industrial uses. The starch content of the leaves, stems, and roots of 2 year-old seedlings is approximately 50 mg·g^−1^ [12], which is equivalent to the starch content of switchgrass (approximately 49.3 mg·g^−1^) [13]. Owing to these advantages, *U. pumila* has great potential for economic and industrial applications. However, soil salinity is a major environmental stressor that restricts the growth and yield of plants, adversely affecting their commercial utilization [14,15]. China has 100 million hm^2^ of salinized land suitable for the cultivation of saline-tolerant bioenergy tree species. *Ulmus pumila* is highly adaptable to adverse conditions, with disease resistance, cold tolerance and drought hardiness, and high tolerance to salt and barren soil [16,17,18,19,20]. It has been reported to be able to tolerate 100–150 mM NaCl (0.3–0.5%), and its growth is not affected at these concentrations [21]. However, the transcriptional regulatory network of *U. pumila* under salt stress has not been adequately explored, and further study is needed to shed light on its salt-tolerance mechanisms at the molecular level. 

Starch is the main component of carbohydrates, and starch biosynthesis is highly sensitive to stresses that affect carbohydrate supply from source organs. Stress-induced growth inhibition is thus related to decreased starch synthesis. Previous studies showed that the starch content of wheat grain was significantly reduced under heat stress [22]. Genes in the starch biosynthetic pathway are jointly regulated under stress, and downregulation of these genes is correlated with starch loss [23]. Previous reports have demonstrated that the activity of enzymes involved in protein and starch synthesis was downregulated in response to shading, resulting in reduced starch deposition; the effects were severe at higher levels of light deprivation [24]. The expression of granule-bound starch synthase I (GBSSI) in rice seedlings declined under drought stress [25]. Some regulators controlling starch biosynthesis genes have been identified. Two proteins, EREBP (ethylene responsive element binding protein) and MYC (v-myc avian myelocytomatosis viral oncogene homolog), may enhance the transcription of the WAXY gene [26]. In the seeds and sink tissues of rice, 307 coexpressed genes are thought to be involved in the regulation of starch biosynthesis, among which APETALA2 (an AP2/EREBP family transcription factor) was found to downregulate the expression of genes involved in starch synthesis [27]. The starch content of *U. pumila* is significantly higher than that of other trees; however, it is not clear whether starch biosynthesis is related to salt tolerance in this species. 

Photosynthesis and carbon fixation are the foundation of energy production in plants and the initial driver of starch synthesis. Photosynthesis is sensitive to various abiotic stresses. The leaves of *U. pumila* are more efficient than the seeds with respect to photosynthesis and electron transfer under low temperatures. The photosynthesis of both leaves and seeds may recover after mild chilling, whereas the photosynthetic system is damaged by high-intensity chilling, eventually leading to death [28]. Many studies focused on the functions of photosynthesis-related genes, aiming to improve photosynthesis and increase harvest. Ribulose-1, 5-bisphosphate carboxylase oxygenase (RuBisCO) is an essential enzyme in the secondary phase of photosynthesis [29]. A decrease in the accumulation of *RuBisCO* transcripts was observed in *Solanum lycopersicum* after diclofenac treatments; reductions in starch content were also observed [30]. A new photorespiratory bypass (GOC bypass) was designed in which no reducing equivalents were produced during the complete oxidation of glycolate into CO_2_ catalyzed by catalase (CAT), glycolate oxidase, and oxalate oxidase. Transgenic, GOC bypass-carrying plants exhibited significant increases in nitrogen content, photosynthetic efficiency, biomass, and yield under both field and greenhouse conditions [31]. Analyzing the adaptive response and recovery capacity of the photosynthetic system of *U. pumila* will clarify the salt tolerance mechanisms in the species. 

We subjected five cultivars of *U. pumila* to four different salt concentrations. We evaluated the threshold concentration at which growth was inhibited by assessing changes in physiology and growth. We also conducted a transcriptome analysis after 3 months of exposure to 150 mM salt stress. Finally, based on the expression patterns of stress-responsive genes, we constructed coexpression regulatory networks for differentially expressed genes (DEGs) under salt stress. Our findings provide new insights into the mechanisms underlying salt stress responses in *U. pumila*, and identify candidate core regulatory factors for further genetic improvements.

## 2. Results

### 2.1. Differences in Salt Tolerance among Cultivars

To identify the salt tolerance limit beyond which growth was inhibited, three different concentrations of NaCl (100 mM, 150 mM, and 200 mM) were applied to 2 year-old seedlings of five different cultivars of *U. pumila* between May and October (i.e., a full growing season). In the 100 mM treatment, the survival rates of the five cultivars did not differ significantly from those of untreated control plants in October (Figure 1a). By contrast, 150 mM NaCl reduced survival rates of Upu2 to 45%, and Upu5 and 12 to 70% in October (Figure 1a,b); however, Upu11 survived completely. When under 200 mM NaCl treatment, the survival rate of Upu11 was decreased by only 10% between June and October (Figure 1c). The Upu11 grew well under 100 mM and 150 mM NaCl treatment in July (Figure 1d). These results indicate that 150 mM NaCl was an appropriate concentration with which to evaluate the salt tolerance of different *U. pumila* cultivars.

We recorded the rates of increase in height and biomass accumulation of seedlings under different salt concentrations, and found that Upu5, 8, 11, and 12 exhibited increased rates of growth and biomass accumulation between May and August under the 0 mM and 100 mM treatments. The height and biomass growth rates of Upu2, 5, 8, 11, and 12 all decreased sharply between August and the end of the growing season. In Upu2, by contrast, the positive trend in growth and biomass accumulation continued until September but declined thereafter. In the 150 mM treatment, the rates of growth and biomass accumulation increased continually in all five cultivars until July; however, increases in height and biomass declined for all cultivars from May onward in the 200 mM treatment. In addition, the increases in the height and biomass growth rates of seedlings in the 100-mM treatment were significantly higher than those of the control plants, whereas they decreased in the 150 mM and 200 mM treatments.

Increases in rates of growth and biomass accumulation were significantly greater in the 150 mM treatment between May and July than in the control group and the other salt treatments, but decreased significantly between August and October. This indicates that 90 days of exposure to 150 mM NaCl is the threshold at which the growth of *U. pumila* becomes inhibited, rather than stimulated. We observed significant differences in tolerance to salt stress among the cultivars (Figure 2), with Upu11 exhibiting higher rates of survival and growth, which is indicative of an enhanced salt-tolerance capacity (Figure 1). We, therefore, used Upu11 in the following analyses.

### 2.2. Physiology and Photosynthesis of Upu11 under Salt Stress

To assess the impacts of salinity, we evaluated select parameters related to salt stress. The contents of soluble sugar (SS), soluble protein (SP), malondialdehyde (MDA), and free proline (PRO) were significantly higher at 150 mM NaCl than under the control conditions (Figure 3a–d). SS and SP contents peaked in July (2.45 ± 0.05 mg/g FW and 21.98 ± 1.06 mg/g FW, respectively), and then decreased significantly, while PRO content exhibited a rapid initial increase that slowed after July. MDA content increased continuously throughout the study period. These results imply that significant increases in SS and SP may be the main drivers of growth in *U. pumila* under salt stress. The net photosynthetic rate (Pn) was significantly higher under the 150 mM NaCl treatment than in the control group between May and July, but decreased significantly thereafter, and was lower than the control group for the last 3 months of the experiment (Figure 3e). This indicates that increased photosynthesis might provide adequate amounts of carbon to maintain growth under salt stress. Stomatal conductance (Gs) decreased significantly under salt stress; however, there were no obvious changes in intercellular CO_2_ concentrations (Ci) (Figure 3f,g), implying that decreases in Ci under salt stress were primarily attributable to non-stomatal limitations. Responses in apparent mesophyll conductance (AMC) indicated that the carboxylation efficiency (CE) of *U*. *pumila* was enhanced during the first 3 months of salt treatment, which might have been the main driver of the observed increases in Pn (Figure 3h). Transpiration rates (Tr) decreased under salt stress, owing to reduced Gs; however, water-use efficiency (WUE) improved between May and July (Figure 3i,j), potentially facilitating dry matter accumulation during the first 3 months of the experiment.

Exposure to salt stress significantly increased both the light saturation point (Lsp) and the light compensation point (Lcp) of Upu11 between May and July, indicating increases in the utilization of both strong and weak light. Meanwhile, increases in dark respiration rates (Rd) indicated that both the energy consumption and metabolic load increased, promoting stress-resistant energy consumption (Figure 3k–n). We also observed increases in the apparent quantum yield of photosynthesis (AQE) between May and July under salt stress, indicating that higher light energy conversion efficiency was one of the factors driving enhanced Pn. After July, the light utilization capacity and efficiency decreased significantly under salt stress. Rd remained unchanged over the same period, resulting in an imbalance in photosynthetic–respiratory energy metabolism. 

Analysis of the CO_2_ response curve showed that the light respiration rates (Rl) of salt-treated plants were higher than those of the control group prior to July (Figure 3o–r); CE also increased significantly between May and July. There were no obvious differences in the CO_2_ saturation point (CSP) between salt-stressed and control plants between May and July; after this point, however, CSP declined significantly until the end of the growing reason, indicating a reduction in photosynthetic capacity. The CO_2_ compensation point (CCP) of salt-treated plants was lower than that of the control group, implying higher carbon assimilation ability, and it was significantly affected by stress. Between August and October, CSP and CE decreased significantly, indicating a decrease in photoreaction capacity and efficiency. The increased CCP indicates reduced carbon assimilation capacity, implying that the plants required more CO_2_ to balance the CO_2_ produced by photosynthesis. At this point, the carbon assimilation capacity weakened. Rl was also significantly reduced, indicating the decreased protection of photoreaction centers.

### 2.3. Functional Annotation and Classification of DEGs

To identify the genes involved in salt stress responses in *U. pumila*, mature leaves from control and salt-treated Upu11 plants were sampled repeatedly over a single day in July for transcriptome sequencing. The collection was conducted at 09:00 (T09), 12:00 (T12), 15:00 (T15), 18:00 (T18), and 21:00 (T21). Samples from control and treat plants were used to prepare cDNA libraries, and libraries were subjected to paired-end sequencing. After filtering low-quality reads, 629.5 million high-quality reads remained, with a mean length of 918 bp and a Q30 > 92.67% (Appendix A). The reference transcriptome was de novo assembled using Trinity. Unique transcripts with an N50 length of 1731 bp were retained in the reference transcriptome, the longest transcript of each gene serving as the unigene. In total, 207,055 unigenes were obtained, with a transcript N50 length of 1896 bp, indicating good quality (Appendix A).

Subsequently, the assembled unigenes were annotated by KEGG ortholog (KO), clusters of orthologous groups of proteins (KOG/COG), NCBI non-redundant nucleotide sequences (NT), NCBI non-redundant protein sequences (NR), protein family databases (PFAM) and the Swiss-Prot database. More than 162,197 unigenes (78.33%) had annotations in at least one database (Appendix A). Of these, 148,896 (71.91%) were annotated in the NR database, and 102,559 (49.53%) were assigned to at least one gene ontology (GO) term. Unigenes were classified into 57 functional groups in the KO database (Appendix A). In total, 63,000 unigenes were allocated to at least one KEGG ontology ID and mapped to 19 pathways (Appendix A).

### 2.4. Identification and Functional Annotation of Salt-Induced DEGs 

To further evaluate DEGs between the control and salt treatments, we used a threshold *Q*-value of <0.005 and a |log2 (fold change)| of >1. We detected 7009 DEGs that were differentially expressed at least at one-time point (Appendix A). In total, 2927 genes were expressed differently at T09 between the control and salt treatment groups (1603 upregulated and 1324 downregulated), 2798 at T12 (1702 upregulated and 1096 downregulated), 3217 at T15 (1560 upregulated and 1657 downregulated), 2691 at T18 (1358 upregulated and 1333 downregulated) and 2482 at T21 (1246 upregulated and 1236 downregulated) (Appendix A). The maximum number of DEGs (3217) were observed at T15, indicating that the treatment had the greatest impact on transcription expression at this time. 

To explore the possible functions of salt-induced DEGs, we assessed DEGs using GO annotation and a KEGG enrichment analysis. DEGs were annotated to 57 GO terms (Figure 4a–c; Appendix A). These DEGs are mostly involved in “photosynthetic membrane”, “PSII oxygen evolving complex”, “photosystem”, “PSII”, and “oxidoreductase complex”, and have implications for various biological processes, including oxidation–reduction, photosynthesis, and light harvesting. The KEGG analysis indicated that these genes are mostly involved in pathways including photosynthesis-antenna proteins, photosynthesis, carbon fixation, hormone signal transduction, and MAPK signaling pathways (Figure 4d; Appendix A). These results demonstrate that photosynthesis and carbon metabolism in *U. pumila* are sensitive to salt stress.

### 2.5. Weighted Gene Correlation Network Analysis (WGCNA) 

To identify hub genes of transcriptional regulation networks responding to salt stress, we conducted weighted gene correlation network analysis (WGCNA). Twelve physiological and photosynthetic traits were included in the analysis. We calculated soft-thresholding powers from 1 to 20 using scale-free topology criteria and used a power of five to identify modules (Figure 5a). The minimodule size was set to 30, and the merge cut height was set to 0.25. As a result, we obtained 17 coexpression modules (Figure 5b), each with 41–3818 genes (Appendix A). Interactions between the coexpression and trait modules were analyzed (Figure 5c); the results indicated that the purple module had the most significant correlation with Pn (r = 0.86, *P* = 6 × 10^−4^). The purple module was significantly positive correlated with Pn and SS modules, with correlation coefficients of 0.68–0.86. The brown module was significantly correlated with the largest number of phenotype trait modules, including Ci, WUE, and SP. These results suggested that DEGs in the purple and brown modules may play more important roles in salt tolerance than those of other modules.

Subsequently, we calculated the connectivity between genes in each module. Genes with high intramodular connectivity were considered intramodular hub genes. Of note, we detected two important hub genes, namely, the ferredoxin reductase gene PHOTOSYNTHETIC ELECTRON TRANSFERH (UpPETH) (purple module), an important electron transporter in the photosynthetic electron transport chain, and UpWAXY (brown module), a key gene controlling the synthesis of amylose in the starch synthesis pathway (Appendix A). The results imply that these genes play an important role in the growth of *U. pumila* under salt stress. The purple module containing UpPETH contained 47 genes and 1 annotated transcription factor, whereas the brown module containing UpWAXY contained 353 genes and 6 annotated transcription factors (Appendix A).

### 2.6. Transcript Regulation in the Photosynthesis and Starch-Synthesis Pathways

We identified 255 salt-responsive genes involved in the photosynthesis pathway, including 153 genes related to the light reaction, 94 related to the Calvin cycle, and 8 related to photorespiration (Figure 6a; Appendix A). With respect to the light reaction, 12 genes were upregulated under salt stress, and 141 were downregulated. In the PSII reaction center, five genes were significantly upregulated, including PHOTOSYSTEM II REACTION CENTER PROTEIN (UpPSB) and the A, E, O, and P genes. In the cytochrome b6/f complex, the PHOTOSYNTHETIC ELECTRON TRANSFER (UpPET) A and PETC genes were upregulated. In the PSI reaction center, the PHOTOSYSTEM I SUBUNIT (UpPSA) F and PSAL genes were significantly upregulated. In redox chain, the hub gene UpPETH was downregulated at T12 under salt stress, but subsequently upregulated. Nine genes related to the Calvin cycle were significantly upregulated, including SUBUNIT OF RuBisCO (UpRBCL), GLYCERALDEHYDE-3-PHOSPHATE DEHYDROGENASE OF PLASTID (UpGAPDH) 1 and 2, GLYCERALDEHYDE 3-PHOSPHATE DEHYDROGENASE A SUBUNIT (UpGAPA) 1 and 2, FRUCTOSE-BISPHOSPHATE ALDOLASE (UpALDO) 1, 2, and 3, and TRANSKETOLASE 1 (UpTKL1). During the photorespiration reaction, the RBCL gene was significantly upregulated in chloroplasts, and two glycine cleavage system H protein (UpGCSH) genes were significantly upregulated in the mitochondria. By contrast, no genes related to photorespiration in peroxisomes were observed to respond to salt stress.

In the starch biosynthesis pathway, 41 genes were differentially expressed under the salt treatments (Figure 6b; Appendix A), of which 12 were upregulated and 29 were downregulated. Three genes related to ALDO biosynthesis were significantly upregulated, whereas the other 19 were significantly downregulated. PHOSPHOGLUCOSE ISOMERASE 1 (UpPGI I), which connects the Calvin cycle to the starch biosynthetic pathway in leaves, was also induced under salt stress. Numerous genes were significantly upregulated in the downstream reaction process under salt stress, including FRUCTOSE-1, 6-BISPHOSPHATASE (UpFBP), two PHOSPHOFRUCTOKINASE 1 (UpPFK1) genes, GLUCOSE-6-PHOSPHATE ISOMERASE (UpGPI), STARCH SYNTHASE (UpSS), and ADP-GLUCOSE PYROPHOSPHORYLASE SMALL SUBUNIT 2 (UpAPS2). UpWAXY, a key gene responsible for amylose synthesis, was also induced by salt. Three UpWAXY genes were significantly upregulated after T15 under salt treatments. The other three UpWAXY genes were downregulated at earlier time points (T09 and T12).

To validate the RNA-seq results between salt-treated and control plants, time-series expression profiles of eight DEGs were selected for quantitative real-time reverse transcription polymerase chain reaction (qRT-PCR) analysis. As shown in Figure 6c, all eight DEGs exhibited consistent changes between RNA-seq and qRT-PCR results, indicating that the sequencing data are credible. 

### 2.7. Hub Transcript Network Construction Based on WGCNA

The networks of the four target modules with edge weights higher than 0.3 were visualized by Cytoscape. Potential regulatory networks with hub genes in target modules are shown in Figure 7a. In the purple module, PETH and its four homologous genes comprise core nodes that interact with the other 25 genes (Figure 7a; Appendix A). The PETH gene encodes ferredoxin NADP^+^ reductase, which is an important electron transporter in the photosynthetic electron transport chain. We overexpressed UpPETH in Arabidopsis and the four overexpression lines (OE) with the highest UpPETH expression levels were selected for further analysis (Appendix A). UpPETH–OE lines exhibited significantly higher survival rates, Pn, biomass, and starch content under salt stress (Figure 7c,d). In core nodes, CHLOROPLAST UNUSUAL POSITIONING 1 (UpCHUP1), CYSTEINE SYNTHASE A (UpCYSK), CELLULOSE SYNTHASE A (UpCESA), INORGANIC PYROPHOSPHATASE (UpPPA), Β-AMYLASE (UpBMY), ABSCISIC ACID RECEPTOR PYL8 (UpPYL8), and DNA-3-METHYLADENINE GLYCOSYLASE II (UpALKA) connect with UpPETH and its four homologous genes, implying that these genes may interact with UpPETH, which, in turn, affects photosynthesis and other related biological processes in *U. pumila* under salt stress. 

The UpWAXY gene, a key gene in amylose synthesis, was induced by salt stress. We overexpressed UpWAXY in Arabidopsis and the four overexpression lines (OE) with the highest *UpWAXY* expression levels were selected for further analysis (Appendix A). UpWAXY–OE lines had significantly higher survival rates, Pn, biomass, and starch content under salt stress (Figure 7c,d). In the brown module (Figure 7b; Appendix A), UpALDO and UpWAXY interacted with up to 62 genes, including UDP-GLUCOSYL TRANSFERASE 73C (UpUGT73C), Bmy, CYTOCHROME P450 82G1 (UpCYP82G1), MAGNESIUM CHELATASE SUBUNIT H (UpCHLH), CYTOCHROME P450 71A3 (UpCYP70A3), E3 UBIQUITIN-PROTEIN LIGASE XERICO (UpXERICO), ZINC FINGER PROTEIN CONSTANS-LIKE 5 (UpCOL5), INOSITOL 3-ALPHA-GALACTOSYLTRANSFERASE (UpGOLS), and LIGHT-HARVESTING COMPLEX II CHLOROPHYLL A/B BINDING PROTEIN 6 (UpLHCB6). These results indicate that starch synthesis may be related to these genes, or is modulated by the same upstream regulatory factors.

## 3. Discussion

### 3.1. Increased Photosynthesis Maintains the Balance between Osmotic Regulation and Lipid Peroxidation

Abiotic stress tolerance is a complex quantitative trait, and significant differences have been observed in stress resistance among species, or even genotypes of the same species [32,33,34]. Survival rates of five *U. pumila* cultivars pointed to significantly different levels of salt tolerance, implying that these cultivars might be suitable for afforestation of soils with different salinity levels. Previous studies showed that a high concentration of NaCl showed a negative effect or damage on plant physiology and growth. However, some others revealed that a low concentration of NaCl could significantly promote plant growth and increase chlorophyll and protein contents and yield [35,36,37]. Sodium is an essential micronutrient element for higher plants, and researchers tried to find the optimal level of salinity for growth. In our study, 100 mM NaCl was associated with increased rates of growth and biomass accumulation in all five cultivars, implying that low salinity has a positive impact on the growth of *U. pumila*. Conversely, higher salinity (150 mM NaCl) resulted in reduced rates of growth and biomass accumulation after July, implying that 150 mM NaCl for three months was the maximum threshold of salt tolerance for all cultivars. Chloride (Cl^–^) is the dominant anion in salt. Glycophytes tolerate only low concentrations of Cl^–^ [38]; the absorption of substantial amounts of Cl^–^ results in ion toxicity to the plant enzyme system, thereby interfering with, or otherwise affecting, various normal metabolic processes, resulting in accelerated protein decomposition and reduced photosynthesis. Consequently, plants experience shortages of nutrients required for growth and development, which in turn reduces cell-growth rates, prevents cell division, and inhibits growth [39,40]. Therefore, the growth inhibition after July may be attributable to the Cl^–^ accumulation in plants due to their prolonged exposure to salinity. The ability of *U. pumila* to survive at 150 mM demonstrates its capacity to tolerate this level of NaCl for three months.

MDA concentration increase due to lipid peroxidation is a good indicator of oxidative stress [41]. PRO, SS, and SP act as excellent osmolytes and are beneficial to plants under various types of stress [42]. The MDA content increased rapidly at 150 mM NaCl, implying higher lipid peroxidation under oxidative stress. Meanwhile, the content of PRO, SS, and SP in Upu11 showed a consistent pattern of change in the first three months of the experiment, indicating an antagonistic relationship between osmotic regulation and membrane lipid peroxidation. The synthesis of numerous osmolytes consumes a substantial amount of the chemical energy produced by photosynthesis. Therefore, increasing photosynthesis may be an important safeguard in maintaining this antagonistic process, and the salt-induced photosynthesis enhancement may be the reason for plant growth promotion. However, photosynthesis was significantly suppressed after July, resulting in decreased energy supplies and reductions in PRO, SS, and SP. These were associated with further increases in MDA content, leading to an imbalance between osmotic regulation and membrane lipid peroxidation, which constrained growth and biomass accumulation.

### 3.2. Mild Salt Stress Induces Expression of Genes Promoting Photosynthesis

Photosynthesis is a complex process involving photosynthetic pigments, photosystems, the electron transport system, and CO_2_ fixation. Among these components, PSII was thought to be the component most sensitive to abiotic stress [43]. We found that 93.1% (27/29) of the genes in the PSI reaction center were suppressed under salt stress, which were higher than those related to the PSII reaction center and electron transport system, implying that the PSI reaction center may have suffered more negative effects. Only two genes, PHOTOSYSTEM I SUBUNIT F (PSAF) and L (PSAL), in the PSI center were induced by salt stress. PSAF is an important subunit of the PSI reaction center, and has been shown to respond to low temperatures and biotic stressors [44,45]. However, our results indicate that UpPSAF was upregulated in *U. pumila* under salt stress. PSAL, along with PSA-H, I, and O, assemble into a peripheral protein on the left side of the PSI complex [46]. The regulation of the stability of PSAL affects the formation of the Light Harvesting Complex I-Light Harvesting Complex II supercomplex [47]. Our results demonstrate that UpPSAL was induced by salt stress, potentially maintaining the formation of the PSI-LHCI-LHCII supercomplex in *U. pumila*.

Repression of the linear electron flow between two reaction centers is the most significant limitation in photosynthesis [48]. Of note, three genes related to the redox chain (UpPETA, UpPETC, and UpPETH) were induced by salt stress. The petc mutant reduces electron transport capacities at saturating light intensities. By contrast, the overexpression of PETC increases electron transport rates and biomass yield [49]. Our results indicate upregulation of the UpPETC gene; this may have ensured electron transport rates in *U. pumila* under salt stress. Previous study reported that increasing levels of PETC resulted in concomitant increases in the levels of PETA and PETB [49]. The peth mutant exhibits permanent induction of photoprotective mechanisms and heavily downregulated photosynthetic machinery [50]. In *U. pumila*, the UpPETH gene was induced under salt stress and was the hub node of the salt-induced transcript regulation network. This indicates that the UpPETH gene might be an important candidate gene for improving photosynthesis in *U. pumila* under salt stress. 

In the PSII reaction center, PSBA, PSBE, PSBO, PSBP, and PSBQ were induced by salt stress, which were proved to be related to the stability of the photosynthetic system. For example, PSBA encodes the chlorophyll binding protein D1, which is dynamically regulated in response to environmental signals [51]. Photosynthesis is inhibited when D1 synthesis plateaus or decreases [52]. PSBE encodes the PSII cytochrome b559, which plays a protective role by acting as an electron acceptor or donor under conditions where the electron flow through PSII is suboptimal [53]. The PSBO gene encodes an extrinsic subunit of PSII that plays a central role in stabilizing the catalytic manganese cluster. Repression of the PSBO gene affects the dephosphorylation of the D1 protein [54]. The PSBQ gene encodes a subunit of the NAD(P)H dehydrogenase complex. The activity of chloroplast NAD(P)H dehydrogenase (NDH) complex in psbq mutants was severely impaired [55]. All these evidences imply the crucial role of these genes in the photosynthesis of *U. pumila* under salt stress. 

In photorespiration, the RBCS gene contributes to the accumulation of RuBisCO in leaves and works additively to yield sufficient RuBisCO for photosynthesis [56]. Our results indicate that the RBCS gene was significantly upregulated under saline conditions, implying it might enhance photosynthetic capacity in response to salt stress. GLYCINE DECARBOXYLASE COMPLEX H (UpGDCH) encodes the glycine decarboxylase complex H protein and is involved in photorespiration. Photorespiration is a carbon recovery system and is catalyzed by ribulose-1, 5-bisphosphate carboxylase/oxygenase [57]. It also links photosynthetic carbon assimilation with nitrogen and sulfur assimilation and C_1_ metabolism, and may contribute to maintaining the redox balance between chloroplasts, peroxisomes, mitochondria, and cytoplasm [58]. Two GDCH genes were found to be upregulated in our study, implying that it might have a positive regulatory role, protecting photosynthesis under conditions of salt stress.

In the Calvin cycle, GLYCERALDEHYDE-3-PHOSPHATE DEHYDROGENASE C-2 (UpGAPC2) encodes glyceraldehyde-3-phosphate dehydrogenase, which interacts with the receptor protein kinase FERONIA (UpFER), which in turn contributes to energy production. Consequently, the available energy is the major factor controlling cell growth. GAPC-2 has been shown to be induced by cadmium [59]. Our results indicate that the expression of UpGAPC2 is also activated by salt stress and might be another positive factor controlling the growth of *U. pumila* under salt stress. FBA encodes fructose 1, 6-biphosphate aldolase, which is involved in gluconeogenesis and glycolysis in the cytoplasm, and Calvin cycle in plastids. In *Arabidopsis*, the fresh weight of fba2 mutant leaves attained only 56% that of the wild type (WT) [60]. AtFBA6 and AtFBA8 have been found to be involved in stress responses and development, respectively. This implies that salt-induced FBA members might be positive regulators of the growth and development of *U. pumila* under salt stress. TK encodes plastid transketolase, which has a significant effect on photosynthesis and plant growth in tobacco [61]. Suppression of TK resulted in a 20–40% decrease in photosynthesis. Increases in transketolase activity would produce thiamine and thiamine pyrophosphate for growth and development [62]. Salt-induced TK might be the basis by which *U. pumila* maintains transketolase activity.

### 3.3. Salt-Induced Genes Accelerate Synthesis and Accumulation of Starch 

Starch is the major carbohydrate storage molecule in plants and is regarded as a key molecule in regulating plant responses to various abiotic stresses. When photosynthesis is limited by adverse environmental conditions, plants generally remobilize starch to supply carbon and energy; the released sugars and other derived metabolites support growth and act as osmoprotectants and compatible solutes to mitigate the negative effects of stress [63]. The GBSS gene encodes the granule-bound glucosyltransferase, which is responsible for elongating amylose polymers [64,65]. The amount of GBSS on starch granules is the main factor driving variations in amylose content [66]. GBSS was found to be induced by salt stress in *U. pumila*, implying that salt stress might promote starch accumulation. STARCH SYNTHASE 3 (SS3) encodes a starch synthase containing a starch-binding domain (SBD), which, in addition to its role in starch biosynthesis, also downregulates the biosynthesis of transient starch [67]. The overexpression of SS3 results in higher levels of both fermentable sugars and hydrolyzed cellulose, as well as increased biomass. Cell-wall properties, such as laxity and degradability, are also affected [68]. Our results show that SS3 was also induced by salt stress in *U. pumila*, implying that salt stress signaling might improve cell-wall properties. Our approach represents a promising biotechnological tool for reducing biomass recalcitrance and the need for pretreatment. 

APS2 encodes the small subunit of ADP-glucose pyrophosphorylase (AGPase), which is an important regulatory component of the starch synthesis machinery [69]. AGPase is activated and inhibited by 3-phosphoglycerate (PGA) and inorganic phosphate (Pi), respectively. The PGA to Pi ratio in the chloroplast stroma directly regulates the activation state of AGPase and is, thus, correlated with leaf starch content. Expression of APS increases under illumination and decreases in the dark, which is consistent with the patterns of starch synthesis in leaves. In the dark, the mostly degraded leaf starch provides glucose skeletons for respiratory processes [70,71]. Our results show that APS2 was induced only at T15, implying that the starch synthesis machinery was promoted at this time. However, the biological significance of this process remains unclear. 

The FBPA gene encodes fructose 1, 6-bisphosphate aldolase, a key enzyme in the Calvin–Benson cycle. In *Arabidopsis*, fbpa mutants are characterized by smaller rosettes and lower photosynthetic rates. Consequently, accumulation of soluble sugars and starch decreases, and SOD activity increases [72]. We found that the UpFBPA gene was induced by salt stress at T15 and T18, implying that it might play important roles in maintaining Pn and reducing photo damage by regulating a wide range of metabolites. The coexpression of GDCH and FBPA may positively regulate the leaf area and biomass [73]. UpGDCH and UpFBPA were induced by salt stress at the same sampling times, implying that salt stress might promote biomass accumulation in *U. pumila*. PGI-I encodes the plastid phospho-glucose (Glc) isomerase [74]. Arabidopsis pgi1-1 mutants accumulate starch in root cap cells and have a starch synthesis deficiency in leaves [75]. They are also significantly delayed under short-day conditions. Our results show that UpPGI1 was significantly upregulated under salt stress, implying that it might ensure sufficient starch biosynthesis in leaves of *U. pumila*.

UpALDO and UpWAXY interacted with up to 62 genes in the brown module of the WGCNA analysis, including UpUGT73C, UpBMV, UpCYP82G1, UpCYP70A3, UpCHLH, UpXERICO, UpCOL5, UpGOLS, and UpLHCB6. These genes are thought to be involved in various biological processes. The UDP-glycosyltransferase (UGT) family of proteins is responsible for transferring sugar to various small molecules and control a series of metabolic processes. AtUGT79B2 and AtUGT79B3 may be strongly induced by a variety of stresses, such as salt, drought, and cold. Transgenic lines exhibiting UGT79B2/B3 overexpression significantly enhanced resistance to salt, drought, and low temperatures, while the ugt79b2/b3 double mutant lines generated by CRISPR-Cas9 exhibited more sensitivity to adverse environmental conditions [76]. CHLH encodes a subunit of Mg-chelatase, and the magnesium chelator H subunit is one of the key enzymes in chlorophyll synthesis. The LHCB family encodes a series of membrane proteins that bind to pigment molecules in PSII and contains a conserved chlorophyll-binding domain. In addition to capturing and transmitting light energy, they are also widely involved in the adjustment and distribution of excitation energy between PSI and PSII, the maintenance of thylakoid membrane structure, photoprotection, and responses to various environmental conditions [77,78,79,80]. This implies that starch synthesis is associated with a series of metabolic processes and photosynthesis, protects the growth of *U. pumila*, and improves adaptation to salt stress.

## 4. Materials and Methods

### 4.1. Plant Materials and Salt Tsreatments

Two year-old *U. pumila* seedlings representing five cultivars (Upu2, 5, 8, 11, and 12), the main cultivars in the saline–alkali soil of North China, were planted in plastic pots (top/bottom diameter 28/19 cm; height 26 cm) with a mixture of sandy soil and vermiculite (1:1 v/v) in a greenhouse (location: 40°0’33” N and 116°20’16” E; day/night temperature 26 ± 2 °C/18 ± 2 °C; relative humidity 50–60%; natural light). After 1 week (15 April 2019), we selected seedlings of each cultivar with similar growth statuses and divided them into four groups. Each group was treated with either 0 mM, 100 mM, 150 mM, or 200 mM of NaCl solution (five cultivars × four salt levels × fifty seedlings per salt level); the NaCl solutions were applied once every 15 days, with a volume of 1.5 L each time.

### 4.2. Growth and Photosynthetic Traits

To assess growth and physiological traits, the height and survival of the seedlings in all treatments, including controls, were recorded on the 15th day of each month between May and October. Three seedlings from each treatment were randomly selected, washed thoroughly with distilled water, and oven-dried at 105 °C for 25 h, after which their dry mass was determined. The growth rate of the plant height and biomass were calculated according to the following formulas: growth rate = (data of current month − data of last month)/data of last month × 100%. Meanwhile, the contents of SP, MDA, PRO, and SS were measured by the biuret method of the protein detection kit (Solarbio^®^ BC3185, Beijing, China), micro malondialdehyde (MDA) assay kit (Solarbio^®^ BC0025, Beijing, China), proline (Pro) content assay kit (Solarbio^®^ BC0295, Beijing, China), and micro plant soluble sugar content assay kit (Solarbio^®^ BC0035, Beijing, China), following the manufacturer’s instructions, respectively. All leaves used in testing were collected from mature plants at a single location.

Physiological traits, Pn, Tr, Gs, and Ci were measured on fully expanded leaves using the LI-6400 portable photosynthesis system (Licor Corp., Lincoln, NE, USA) on the first sunny day (10:00–11:30) after the 15th day of every month. The photosynthetically active radiation (PAR) was set at 1200 μmol·m^−1^·s^−1^ in the leaf chamber by a LED blue/red light source (Li-Cor), and the ambient CO_2_ concentration was maintained at 380 ± 3 μmol·mol^−1^. Meanwhile, the same portable photosynthesis system was used to measure light-response and Pn/Ci curves. The light-response curves were determined under the condition of 380 μL·L^−1^ CO_2_ supply at 25 °C. The non-rectangular hyperbola was used to calculate the Rd, AQE, Lcp and Lsp of the response of leaf Pn to PAR [81]. WUE and AMC were calculated according to the following formulas: WUE = Pn/Tr [82] and AMC = Pn/Ci [83].

The rectangular hyperbola was used to calculate the CE, Rl, CCP, and CSP of the response of leaf Pn to Ci [84].

Mature leaves were collected along with a time series on a single day, at 09:00, 12:00, 15:00, 18:00, and 21:00. Three biological replicates were collected from the same position at each collection time, immediately frozen in liquid nitrogen, and then stored at −80 °C.

### 4.3. RNA Isolation, Library Preparation, and RNA-Seq

Leaves were collected in July from three biological replicates from both control and salt-treated Upu11 plants, for use in strand-specific transcriptome sequencing. Total RNA was extracted using a Qiagen RNeasy kit (Qiagen, Shanghai, China) following the manufacturer’s protocols, and purified using an RNeasy Plant Mini Kit (Qiagen, Shanghai, China). RNA yield and purity were examined using a nanophotometer spectrophotometer (Implen, Munich, Germany) and agarose gel electrophoresis. Strand-specific sequencing libraries were generated using a NEBNext Ultra Directional RNA Library Prep Kit (New England Biolabs, Ipswich, MA, USA), and sequenced on an Illumina Hiseq platform.

### 4.4. Functional Annotation of Unigenes 

After filtering low-quality reads and assembling the transcriptome, we conducted a functional annotation of the unigenes based on the following databases: KOG/COG (http://www.ncbi.nlm.nih.gov/COG/ accessed on 5 April 2021), KO (http://www.genome.jp/kegg/ accessed on 5 April 2021), NR (ncbi.nlm.nih.gov/blast/db/FASTA/ accessed on 5 April 2021), NT (ncbi.nlm.nih.gov/blast/db/FASTA/ accessed on 5 April 2021), PFAM (http://pfam.xfam.org accessed on 5 April 2021), and Swiss-Prot (http://www.ebi.ac.uk/uniprot/ accessed on 5 April 2021) databases.

### 4.5. Gene Expression and Annotation

To determine the gene expression levels, clean data were mapped to the assembled transcriptome, then the read count of a specific gene was calculated according to the mapping results by RSEM software (v1.3.3) [85]. Genes with an adjusted *p*-value of <0.05 were considered differentially expressed. To identify statistically significant DEGs, we established thresholds based on a Q-value of <0.005 and a |log (fold change)| of >1. We conducted GO enrichment analysis of DEGs with the GOseq package (v1.42.0) in R software (v3.4.4), using Wallenius’ non-central hypergeometric distribution [86], which can accommodate gene length biases in DEGs.

### 4.6. qRT-PCR

We selected expression profiles of eight DEGs for use in qRT-PCR to validate the RNA-seq results. Primer premier v6.0 was used to design gene-specific primers (Appendix A). Total RNAs were isolated from frozen leaf samples collected from both the NaCl and control groups and were reverse transcribed to cDNA using a PrimeScript RT Reagent Kit +rDNA Eraser (TaKaRa, Kusatsu, Shiga, Japan). The resulting cDNAs were then subjected to qRT-PCR analysis using a QuantiNova SYBR Green PCR Kit (Qiagen, Shanghai, China) in a qPCR System (M×3005P, Agilent Technologies, Santa Clara, CA, USA). The results were normalized based on the amount of 18s rRNA in each sample.

### 4.7. Weighted Gene Correlation Network Analysis

We identified hub genes and constructed DEG networks using the time-series samples, using WGCNA with standard methods [87,88] to detect specific modules of coexpressed genes associated with each physiological trait. The expression data of 7009 genes differentially expressed at least one-time point were used as the input expression data. Physiological traits included Pn, Gs, Ci, Tr, WUE, SS, SP, PRO and MDA. The gene-expression matrix was derived from the gene expression at different sampling times. In cases of duplicate sampling, the mean value was used. R software (v3.4.4) and the WGCNA package (v1.6.6) were used to construct a coexpression network and partition related modules. The hub genes of each module were identified based on the connectivity of the genes within each module. The resulting modules were visualized as graph networks in Cytoscape 3.5.1 (http://cytoscape.org accessed on 5 April 2021) using a circular layout to visualize interconnections identified in the datasets. 

### 4.8. Overexpression of Salt-Responsive Hub Genes in Arabidopsis

To construct the overexpression vector for candidate hub genes, coding sequences of UpPETH and UpWAXY from Upu11 were amplified using gene-specific primers (Appendix A) and inserted into pCXSN vector to generate the 35S::UpPETH and 35S::UpWAXY constructs. *Agrobacterium* GV3101 carrying recombinant vector was used to transform *A. thaliana* (Col-0) using the modified floral dip method. Heterologous OE lines (T3) were used for subsequent determinations.

To evaluate the salt tolerance of the transgenic Arabidopsis plants, seeds from transgenic lines harboring UpPETH and UpWAXY, and ecotype Col-0 seeds were surface sterilized and vernalized in the dark at 4 °C for 2 d, plated on 1/2 MS medium containing 0 mM or 150 mM NaCl, and incubated under controlled conditions (22 °C, 16 h/18 °C, 8 h) for further observation. 

### 4.9. Statistical Analysis 

All measurements in this study were carried out in at least three biological and three technique replicates. The statistical analysis was conducted using SPSS (v22.0). *p* < 0.05(*) indicates significant difference between mean values. The data are presented as the mean ± standard deviation (SD) of three independent experiments.

## 5. Conclusions

We identified two core regulatory nodes in *U. pumila* plants exposed to long-term salt stress. Using analyses of transcript regulation pathways, combined with physiological data, we explored the molecular mechanisms of responses to chronic salt stress in *U. pumila* at the transcriptional and physiological levels. Plants acclimatized to salt stress by promoting photosynthetic efficiency (induction of Pn, AMC, WUE, LSP, LCP, and R1, and reductions in Gs), accumulating osmotic regulation substances (SS, SP, and PRO), and accelerating the synthesis and accumulation of starch; these may be the reasons why low salt concentration promotes plant growth. The coexpression network indicated that UpPETH and UpWAXY are core nodes for salt-responsive transcript regulation. Overexpression of UpPETH and UpWAXY significantly increased the survival rates, net photosynthesis rates, biomass, and starch content of transgenic *Arabidopsis* plants under salt stress. These results reveal previously unknown transcriptional photosynthesis and starch metabolism responses in *U. pumila* under salt stress. The genes and pathways that we identified may provide targets for further genetic improvement.

## Figures and Tables

**Figure 1 ijms-22-04410-f001:**
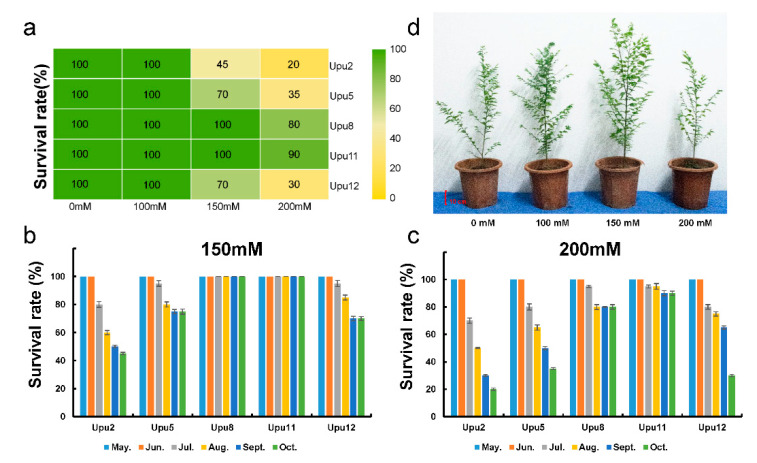
Survival rates of *U. pumila* under salt stress treatment. (**a**) Survival rates of five *U. pumila* cultivars under different concentrations of NaCl in October; (**b**,**c**) survival rates of five cultivars under 150 mM and 200 mM NaCl treatments. (**d**) Phenotypes of Upu11 under NaCl treatments in July. Scale bar 10 cm; data are presented as means ± SD (*n* = 3 [3 groups × 10 plants per cultivar]).

**Figure 2 ijms-22-04410-f002:**
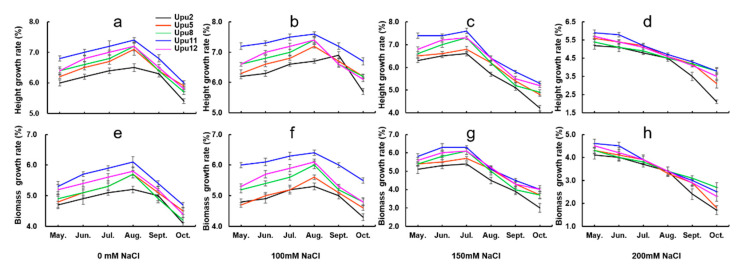
Changes in rates of growth and biomass accumulation. (**a**–**d**) Changes in growth rates from May to October under control and NaCl treatments; (**e**–**h**) changes in biomass accumulation rates from May to October under control and NaCl treatments. Values represent means ± SD (n = 3).

**Figure 3 ijms-22-04410-f003:**
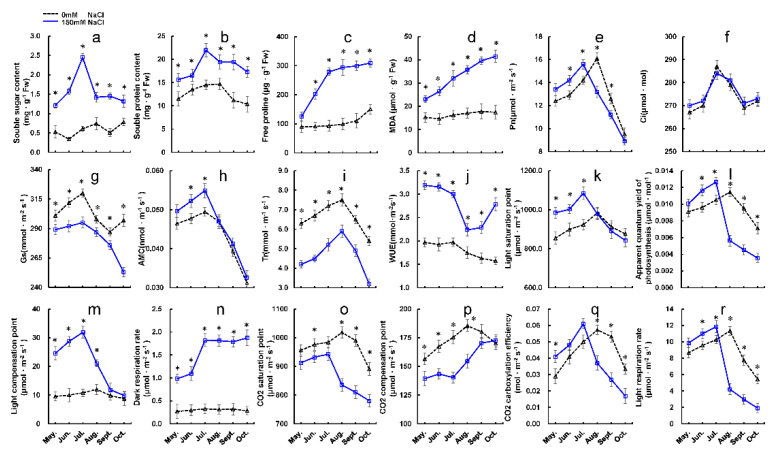
Physiological and photosynthetic changes in Upu11 under 0 mM and 150 mM NaCl treatments. (**a**–**d**) Changes in physiological parameters of protective enzymes, including (**a**) SS, (**b**) SP, (**c**) PRO, and (**d**) MDA; (**e**–**j**) changes in photosynthetic parameters, including (**e**) Pn, (**f**) Ci, (**g**) Gs, (**h**) AMC, (**i**) Tr, and (**j**) WUE; (**k**–**n**) changes in light response parameters, including (**k**) Lsp, (**l**) AQE, (**m**) Lcp, and (**n**) Rd; (**o**–**r**) changes in photosynthetic CO_2_ response curve parameters, including (**o**) CSP, (**p**) CCP, (**q**) CE, and (**r**) Rl. Values represent means ± SD (n = 3), asterisks indicate *p* < 0.05.

**Figure 4 ijms-22-04410-f004:**
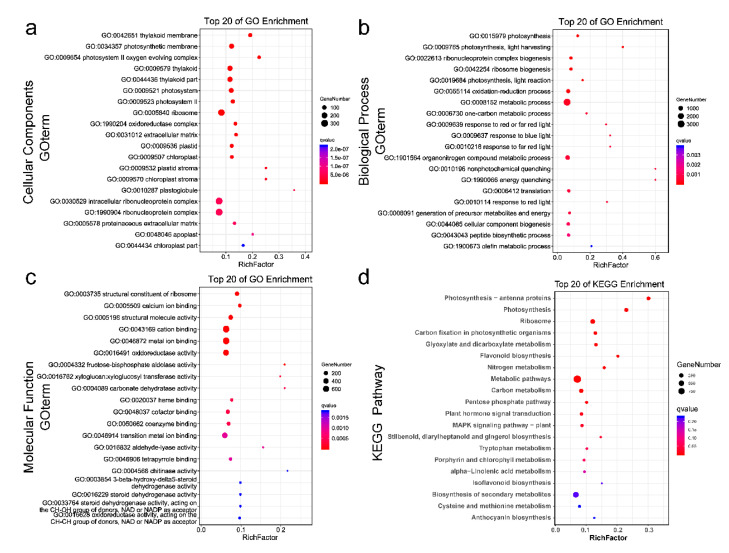
GO annotation and KEGG pathway analysis. (**a**–**c**) Top 20 GO enrichment classifications of DEGs in (**a**) cellular components, (**b**) biological process, and (**c**) molecular function; (**d**) top 20 KEGG enrichment pathways of DEGs. The y-axis represents the top enriched GO or KEGG pathways’ terms. The size and color of circles indicate the number of genes and Q-values of the enriched term, respectively. The x-axis represents the rich factor (RF), which refers to the ratio between the number of DEGs and the number of genes annotated in a given term or pathway. A large RF indicates a high degree of enrichment.

**Figure 5 ijms-22-04410-f005:**
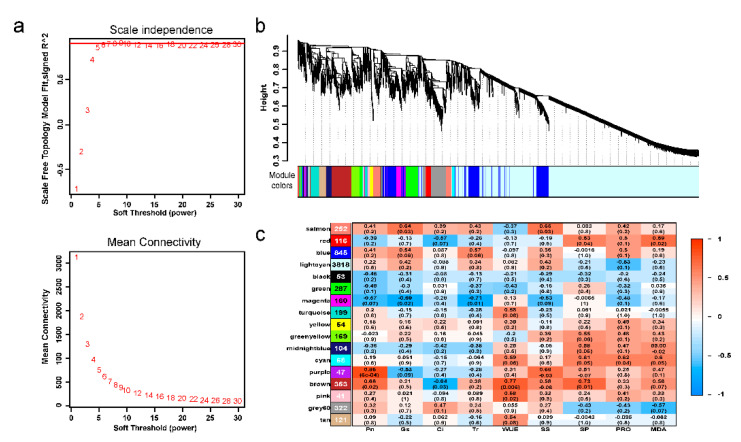
Construction of the WGCNA model. (**a**) Screening of soft threshold powers by network topology analysis. (**b**) Coexpression modules identified by WGCNA. The major tree branches represent 17 modules. (**c**) Module–trait correlation relationships. The number in the first row of each cell represents the corresponding correlation, and the number in parentheses represents the *p*-value. The color of each cell represents the correlation coefficient. The numbers in the leftmost modules represent the numbers of DEGs in this module.

**Figure 6 ijms-22-04410-f006:**
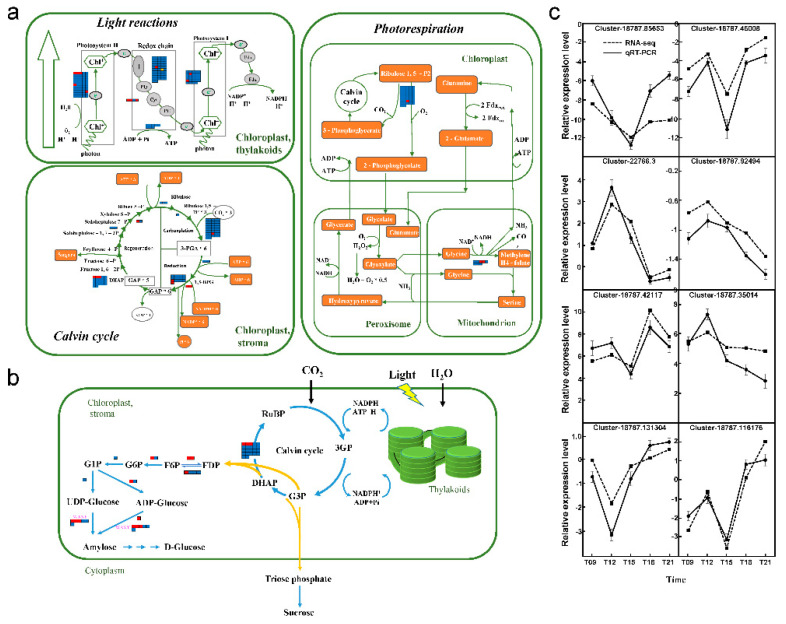
DEGs are involved in photosynthesis, carbon fixation, and starch biosynthesis. (**a**) Photosynthesis and carbon fixation pathway; (**b**) Starch biosynthesis pathway. The color of the rectangle represents the changes of gene expression under NaCl treatment, red indicates upregulated and blue indicates downregulated over the full-time series (09:00–21:00). The yellow represents DEGs with both differentially upregulated and downregulated expressions over the time series. FDP: beta-D-fructose 1, 6-bisphosphate, F6P: beta-D-fructose 6-phosphate, G6P: alpha-D-glucose 6-phosphate, G1P: D-glucose 1-phosphate. The details of gene annotation and expression changes are shown in Appendix A, respectively. (**c**) Verification and expression analysis of selected DEGs by quantitative real-time PCR (qRT-PCR). Data are presented as means ± SD (n = 3).

**Figure 7 ijms-22-04410-f007:**
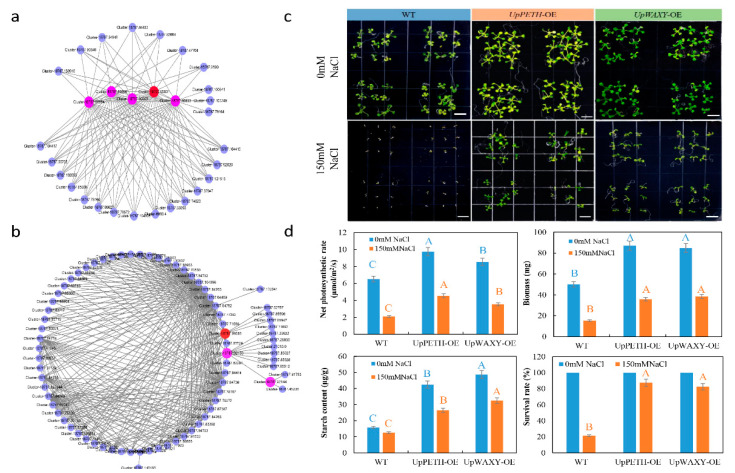
Potential regulatory networks with hub genes in target modules. (**a**,**b**) Gene regulatory network with potential interactions with (**a**) UpPETH (Cluster-18787.85831), and (**b**) UpWAXY (Cluster-18787.99586), respectively. Red nodes represent hub genes (UpPETH or UpWAXY), pink nodes represent genes homologous with the hub gene, and blue nodes represent genes with potential interactions with the hub gene. Details of gene annotation are provided in Appendix A. (**c**) Phenotypes of transgenic and WT Arabidopsis lines with UpPETH and UpWAXY overexpression after 7 days of growth under 0 mM and 150 mM NaCl treatments; (**d**) Pn, biomass, starch content, and survival rates of transgenic plants after 45 days of growth under 0 mM and 150 mM NaCl treatments. Values represent means ± SD (n = 3). Bars with blue uppercase letters indicate statistical significance at *p* <0.01 under the 0 mM treatment. Bars with orange uppercase letters indicate statistical significance at *p* < 0.01 under the 150 mM treatment.

## Data Availability

All transcriptome expression data have been deposited in the Genome Sequence Archive in the BIG Data Center (BIG, CAS, China) under accession numbers CRA003513 (https://bigd.big.ac.cn/ accessed on 5 April 2021).

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
