# Peer review of "Gene Coexpression Network Analysis Indicates that Hub Genes Related to Photosynthesis and Starch Synthesis Modulate Salt Stress Tolerance in Ulmus pumila"

_ijms, 2021, doi:10.3390/ijms22094410_

Round 1
Reviewer 1 Report
The study presented in the manuscript refers to the physiologic and transcriptomic analysis of 5 cultivars of Ulmus pumila L. after 3 months of continuous treatment with 150 mM NaCl. The study aimed to identify genes underlying salt stress responses in U. pumila. The authors conducted a comparison of RNA-seq results and identified DEGs. Classically like in this type of research, they made also GO, KEGG and coexpression network analysis. As the result, the authors proposed some genes that could be related to the salt stress responses. They also performed additional analyzes: physiological and mutants of A. thaliana to complete the picture of Ulmus development in salinized land conditions. This is the first report of this type of research on this plant species. I believe that the results could shed light on previously unknown photosynthesis and starch metabolism responses in U. pumila under salt stress and benefit for future improvements in the breeding of elm.
The work has a typical layout for this type of research. The authors undoubtedly put a lot of effort into research. Generally, the whole manuscript is good written, clear but due to too much text, it is quite difficult to follow. Both the introduction and results and discussion can be shortened without losing quality. I suggest shortening the text and removing some general information about photosynthesis, the described genes, etc., which extend the text and are not necessary to describe the processes in question.
Author Response
Responses to Reviewer 1
We sincerely appreciate the reviewer’s specific suggestions and forward-looking comments, which have helped us to improve the manuscript. Following your valuable comments, we have made our best effort at improving the quality of the manuscript. Thank you for all your time. We have incorporated most of the changes in this revised version.
Specific comments:
The work has a typical layout for this type of research. The authors undoubtedly put a lot of effort into research. Generally, the whole manuscript is good written, clear but due to too much text, it is quite difficult to follow. Both the introduction and results and discussion can be shortened without losing quality. I suggest shortening the text and removing some general information about photosynthesis, the described genes, etc., which extend the text and are not necessary to describe the processes in question.
Response: Thank you for your specific suggestion, we have removed some unimportant descriptive sentences from the introduction, results, and discussion. All deletions were marked in the revised version, for instances:
‘medicine, and the production of high-quality furniture, decorative articles’ ‘The young seeds and leaves, which are rich in starch, dietary fiber, protein, vitamins, and various trace elements, can be used as food or animal feed.’ in the Introduction.
‘We then compared DEGs between adjacent sampling times to determine which was most strongly affected by salt treatments.’ ‘cellular components as ribosome’ ‘regulation, metabolism, organonitrogen compound metabolism’ ‘These DEGs are primarily involved in cation binding, metal-ion binding, oxidoreductase activity, transition-metal-ion binding, and fructose-bisphosphate aldolase activity.’ In the KEGG analysis, all DEGs were successfully classified into five categories, including 130 pathways.’ in Results.
‘However, our results indicate that UpPSAF ...PSAL plays a role in stabilizing PSAH and PSAO.’‘PETA encodes the cytochrome f apoprotein,…’in Discussion.
Reviewer 2 Report
In this manuscript, the author did a Gene co-expression network analysis that indicates hub genes related to photosynthesis and starch synthesis modulate salt stress tolerance in Ulmus pumila.
In this study, the authors exposed five main cultivars in saline-alkali land (Upu2, 5, 8, 11, and 12) to NaCl stress. Of the five cultivars, Upu11 exhibited the highest salt resistance. Growth and biomass accumulation in Upu11 were promoted under low salt concentrations (< 150 mM). However, after three months of continuous treatment with 150 mM NaCl, growth was inhibited, and photosynthesis declined. Transcriptome analysis conducted after 3 months of treatment detected 7,009 differentially expressed unigenes (DEGs). Gene annotation indicated that these DEGs were mainly related to photosynthesis and carbon metabolism. Furthermore, PHOTOSYNTHETIC ELECTRON TRANSFERH (UpPETH), an important electron transporter in the photosynthetic electron transport chain, and UpWAXY, a key gene controlling amylose synthesis in the starch synthesis pathway, were identified as hub genes in the gene co-expression network. They identified 25 and 62 unigenes that may interact with PETH and WAXY, respectively. Overexpression of UpPETH and UpWAXY significantly increased survival rates, net photosynthetic rates, biomass, and starch content of transgenic Arabidopsis plants under salt stress. I have few major concerns related to this study.
Major:
- Place Figure S3 to main Figure 6.
- Figures 1a and 1d don’t match. Upu11 under 150 mM NaCl treatments looks big and better than 0 mM and 100 mM NaCl. How does the author explain this? In figure 1, how many plants were used? Please put n=? in figure legends.
- It is very good that the author tried to validate the function of selected genes, but how many Ox lines were generated in this study. One is not enough to validate any function. There is no information on Ox line validation like genomic DNA PCR RT-PCR validation. How did the author found homozygosity? In T2, it is difficult to get HO lines.
- In figure 7C, control is not equal. In 0mM, NaCl WT shows a week phenotype in the lower panel, which might result in getting a bad phenotype in NaCl because control is not healthy.
Minor:
Changes at
L31 with with to with.
L120 and Upu11 growth well to and Upu11 grew well.
L238, L690 one time point to one-time point.
L338 DEGs involved in photosynthesis to DEGs are involved in photosynthesis.
L331 full time to full-time.
L493 dehydrogenease to dehydrogenase.
L573 fructose 1, 6-bisphophate aldolase to fructose 1, 6-bisphosphate aldolase.
L617 four salt leves to four salt leaves?
L650 along a time to along with a time.
I have found plagiarism at L86-87, L91-93, L221-223, L254, L257-258, L276-278, L287-289, L289-294, L440-441, L461-462, L467-468, L471-472, L493-494, L496-496, L498-501, L522-524, L531-532, L539-541, L546-547, L550-551, L637-644, L665, L668-671, L705-712. Please clean it.
Author Response
Responses to Reviewer 2
We appreciate the reviewer’s specific suggestions and forward-looking comments. Following your valuable comments, we have added some details into the manuscript. We have carefully re-checked and revised all Writing errors in our manuscript.
Q1: Place Figure S3 to main Figure 6.
Response: We thank the reviewer for the detailed comments and we have placed Figure S3 to main Figure 6.
Q2: Figures 1a and 1d don’t match. Upu11 under 150 mM NaCl treatments looks big and better than 0 mM and 100 mM NaCl. How does the author explain this? In figure 1, how many plants were used? Please put n=? in figure legends.
Response: We have revised the description words for Figures 1a and 1d to increase readability.
In addition, We added some sentences to explain why Upu11 under 150 mM NaCl treatments looks big and better than 0 mM and 100 mM NaCl. “However, some others reveled that low concentration of NaCl could significantly promote plant growth, increase chlorophyll and protein contents and yield [35-37].” and “Therefore, increasing photosynthesis may be an important safeguard in maintaining this antagonistic process and the salt-induced photosynthesis enhancement may be the reason of plant growth promotion.”in 3.1. Salt treatment has two different effects on plant growth,low concentration promotes growth, while concentration inhibits growth. 100 mM NaCl, and 150 mM NaCl in the first three months (May, Jun, July) was associated with increased rates of growth and biomass accumulation in all five cultivars, implying that low salinity has a positive impact on the growth of U. pumila. Conversely, Continuous salt treatment with 150 mM NaCl resulted in reduced rates of growth and biomass accumulation after July, implying that 150 mM NaCl for three months was the maximum threshold of salt tolerance for all cultivars. At this time, the promoting effect of salt on plant growth turned into inhibiting effect. Therefore, Upu11 under 150 mM NaCl treatments looks big and better than 0 mM and 100 mM NaCl.
We used 3 groups ×10 plants per cultivar per salt level to calculate the mean and standard deviation of the survival rate. We have added the detail information to the figure legend.
Q3: It is very good that the author tried to validate the function of selected genes, but how many Ox lines were generated in this study. One is not enough to validate any function. There is no information on Ox line validation like genomic DNA PCR RT-PCR validation. How did the author found homozygosity? In T2, it is difficult to get HO lines.
Response: In our study, we obtained nine UpPETH and eleven UpWAXY Heterologous OE lines in T3 respective. Heterologous OE lines (T3) were used for subsequent determinations, including RT-PCR (Figure S3) and salt tolerance evaluation (Figure 7c, d). We have added some details about genomic DNA PCR, RT-PCR validation in section 2.7 in Results and 4.8 in Materials and Methods. The gene-specific primers and protein coding sequences of UpPETH and UpWAXY were showed in Table S15 and S16.
Q4: In figure 7C, control is not equal. In 0mM, NaCl WT shows a week phenotype in the lower panel, which might result in getting a bad phenotype in NaCl because control is not healthy.
Response: We are very sorry for our negligence. We have supplemented the correct figure in this result.
Additionally, we thank the reviewer for the detailed comments and we have revised all the minor writing errors in our manuscript, and we have cleaned and revised the contents that may involve plagiarism.
Round 2
Reviewer 2 Report
Manuscripts have improved a lot. The author mentioned in the reply that they have replaced figure 7c but I didn't see any changes. Once the author changes the figure 7c manuscript can be accepted.
Author Response
Reviewer2:“The author mentioned in the reply that they have replaced figure 7c but I didn't see any changes. Once the author changes the figure 7c manuscript can be accepted.”
Our response: we appreciate this suggesttion. I am so sorry for this error. New figure7c has been added in current manuscript. Please checked it .